# First Gallium and Indium Crystal Structures of Curcuminoid Homoleptic Complexes: All-Different Ligand Stereochemistry and Cytotoxic Potential

**DOI:** 10.3390/ijms242216324

**Published:** 2023-11-15

**Authors:** William Meza-Morales, Yair Alvarez-Ricardo, Leidys L. Pérez-González, Rosario Tavera-Hernández, María Teresa Ramírez-Apan, Rubén A. Toscano, Rubén Sánchez-Obregón, Marco A. Obregón-Mendoza, Raúl G. Enríquez

**Affiliations:** Instituto de Química, Universidad Nacional Autónoma de México, Ciudad de México 04510, Mexico; william.meza@upr.edu (W.M.-M.); yfar30@hotmail.com (Y.A.-R.); leidyslaura92@gmail.com (L.L.P.-G.); rosario.tavera@gmail.com (R.T.-H.); mtrapan@yahoo.com.mx (M.T.R.-A.); toscano@unam.mx (R.A.T.); rubens@unam.mx (R.S.-O.)

**Keywords:** curcuminoids, 3,4-dimethoxycurcumin, homoleptic complexes, gallium complexes, indium complexes, X-ray structure, single crystal

## Abstract

The crystal structure determination of metal complexes of curcuminoids is a relevant topic to assess their unequivocal molecular structure. We report herein the first two X-ray crystal structures of homoleptic metal complexes of a curcuminoid, namely Dimethoxycurcumin (DiMeOC), with gallium and indium. Such successful achievement can be attributed to the suppression of interactions from the phenolic groups, which favor an appropriate molecular setup, rendering Dimethoxycurcumin gallium ((DiMeOC)_2_-Ga) and Dimethoxycurcumin indium ((DiMeOC)_3_-In) crystals. Surprisingly, the conformation of ligands in the crystal structures shows differences in each metal complex. Thus, the ligands in the (DiMeOC)_2_-Ga complex show two different conformers in the two molecules of the asymmetric unit. However, the ligands in the (DiMeOC)_3_-In complex exhibit three different conformations within the same molecule of the asymmetric unit, constituting the first such case described for an ML_3_ complex. The cytotoxic activity of the (DiMeOC)_2_-Ga complex is 4-fold higher than cisplatin against the K562 cell line and has comparable activity towards U251 and PC-3 cell lines. Interestingly, this complex exhibit three times lesser toxicity than cisplatin and even slightly lesser cytotoxicity than curcumin itself.

## 1. Introduction

Gallium (Ga) and indium (In) are widely utilized in various applications [1], including violet lasers, photodetectors, UV cameras, and transistor devices [2]. Additionally, these elements find application in LEDs [3,4]. Over the past two decades, there has been considerable focus on the chemical vapor deposition (CVD) of group 13 metal oxides, namely M_2_O_3_ (M = Al, Ga, and In) [5,6]. However, exploration of their applications in the field of biology and medicine has been limited.

Gallium (III) complexes have attracted attention due to their potential biological properties, as previously reported [2]. While gallium nitrate has received FDA approval for cancer treatments [7], reports on the biological applications of gallium complexes remain scarce. Although Ga^3+^ and In^3+^ are not considered essential ions, their similarities to essential ions like Fe^3+^ or Ca^2+^ enable them to participate in biochemical processes in vivo [8]. A recent investigation involving Ga(NO_3_)_3_ and In(NO_3_)_3_ demonstrated their potent antineoplastic activity (in vivo), resulting in significant growth inhibition of various solid tumors with low toxicity [7]. Additionally, complexes of curcumin and curcuminoids with indium and gallium have found applications as radiotracers [9,10,11,12].

In previous studies, complexes of curcumin and other curcuminoids with gallium and indium have reported the ML_3_ metal-ligand relationship using various spectroscopic techniques [9,10,11,12]. We placed our interest in obtaining crystal structures that would enable the unambiguous determination of the metal:ligand relationship. This was achieved by the use of the single crystal X-ray (SCXR) technique to eliminate stoichiometric uncertainties.

We address the fact that a common source of uncertainty when dealing with curcuminoid complexes of gallium and indium is stoichiometry. A possible explanation can be found in the significant differences in ionic radii between gallium and indium (ca. 30 pm), which makes it difficult for both metals to form the expected ML_3_ complexes with the curcuminoid scaffold. It is known that the coordination number is influenced by ionic radii. Thus, in the case of Ga^3+^, the ionic radii range from 47 pm to 62 pm, resulting in variable coordination numbers such as tetracoordinate, pentacoordinate, and hexacoordinate. On the other hand, In^3+^ has ionic radii ranging from 62 pm to 92 pm, leading to tetracoordinate, hexacoordinate, and octacoordinated complexes [13]. In our previous work [14,15,16], we have found that the X-ray structure determinations are critical for establishing the stoichiometric relationships to the metal, geometry, and the conformations of ligands. The curcumin-copper complex recently reported provides a suitable ML_2_ example [14]. The curcuminoid complexes of Zn we recently reported provide an example of different geometries, including octahedral, trigonal pyramidal, and trigonal bipyramidal [15]. The heterocyclic curcuminoid complexes with Mg and Cu provide an example of the first conformationally heteroleptic complexes [16]. To the best of our knowledge, no theoretical approach has predicted any of the experimental cases mentioned. On the other hand, the use of non-curcuminoid auxiliaries (such as phenantroline [17,18], bipyridine [19], acetylacetone, etc) to promote crystallization implies a departure in the attempt to improve the biological activity of curcuminoid ligands linked to a metal. In such cases, the resulting complex has only a partial curcuminoid nature.

The investigation of Group 13 (Al, Ga, In, Tl) concerning metal cluster compounds [2], and ligands such as β-diketiminate [20] or acetylacetonates (ACAC), has been extensively used as supporting ligands, conferring protection to prevent secondary reactions or nucleophilic attacks on the metal ion [21]. Therefore, the chemistry of organogallium and organoindium compounds using other ligands such as curcumin or curcuminoids widens the knowledge of this type of metallodrug.

The selection of an appropriate ligand plays a crucial role in designing a metallodrug that is both chemically stable and biologically active [22]. Curcumin is a natural product that belongs to the polyphenols group, present in *Curcuma longa* [23], with known anti-inflammatory, antioxidant, anticancer, antibacterial and antiviral activity [24], which include several types of clinical trials [25]. It is noteworthy to mention that the dimethoxy derivative of curcumin, is not a naturally occurring metabolite, yet it exhibits high biological activity [26,27].

In this investigation, we explored the metal complexation of dimethoxycurcumin, which contains the β-diketone system of curcumin and resembles that of acetylacetone (ACAC). Furthermore, we developed appropriate conditions for the synthesis of gallium and indium metal complexes using salts of both ions. The principal focus of this study is to examine the relationship between the ligand and metal coordination number, primarily through solid-state analysis. This was achieved by means of monocrystal X-ray diffraction, allowing the elucidation of the chemical structures of both gallium and indium complexes. A secondary goal was the assessment of the cytotoxic activity of both complexes against cancer cell lines.

## 2. Results

Dimethoxycurcumin (DiMeOC) **1** was obtained in a 63% yield and used as a ligand to form the gallium (81% yield) and indium (66% yield) complexes (Figure 1). They were further characterized through single crystal X-ray diffraction, solid and liquid state NMR, UV-Vis, fluorescence, IR, and HRMS techniques along with the evaluation of their cytotoxic activity.

## 3. Discussion

### 3.1. DRX Single Crystal

(DiMeOC)_2_-Ga (**2**). The crystal structure consists of two distinct ionic fragments occupying different crystallographic inversion centers (Figure 2). Each cation is formed by a gallium atom surrounded by two DiMeOC ligands and two molecules of DMF in a slightly elongated octahedral coordination geometry. The two chelating DiMeOC ligands define the symmetry-generated equatorial plane of the octahedron, while the two molecules of DMF lie trans to each other on the vertical axis. The anion is a rotationally disordered tetrachlorogallate**.** The overall structure closely resembles the structure of [*trans*-Ga(acac)_2_(THF)_2_^+^][GaCl_4_^−^] reported by Beachley et al. [28].

(DiMeOC)_3_-In (**3**). The three DiMeOC ligands are arranged in a slightly compressed (O1, O3) octahedral polyhedron. The range of In–O bond lengths 2.1156 (13)–2.1435 (13) Å was consistent with previously reported homoleptic indium β-diketonates, but the ligand bite in the range 2.917–2.950 Å is significantly wider. Nevertheless, despite several crystal structures of homoleptic tris (β-diketonates) of trivalent metals being reported [28], this is just the second example of a tris (curcuminoid) complex.

The most striking feature of both complexes is the occurrence of several conformations of the ligand DiMeOC trapped in these complexes.

Assuming the *s-trans* (C1, C2, C3, C4), *s-cis* (C3, C4, C5, O2), *s-cis* (O2, C5, C6, C7) keto-enol tautomer as the most stable conformation of the (*E*,*E*)-1,6-heptadiene-3,5-dione in DiMeOC, the ligand still can adopt three main conformations (cis-up, cis-down, and trans, Figure 3) depending on the relative orientations of the 3,4-dimethoxyphenyl and the β-diketonate groups. A conformer search performed at the molecular mechanics level with the MMFF94s force field using the GB/SA ethyl acetate continuum solvent model as implemented in CONFLEX [29] showed that at 298 K, the free energy difference among the main conformers (94.6983% of the population) is less than 1.50 kcal/mol, with the relative populations reflecting the energy difference between the different conformations (Appendix A). Two of these selected conformations (“cis-down” and “trans”) were found in the crystal structures of two polymorphs of the ligand DiMeOC [30].

Heteroleptic and homoleptic coordination compounds with this ligand also display individual examples of the “cis-down” [15,31] and “trans” conformations [32,33].

Most examples of crystal structures involving several conformers are polymorphs, products of phase transitions, or disordered crystal structures. Interestingly, when comparing the structure of both cations in compound **2**, the main difference between them involves the conformation adopted by the DiMeOC ligands: “cis-down” in Ga1 and “trans” in Ga2 cationic complexes (Figure 2). Furthermore, the co-existence within the same crystals of two conformations for the chelate rings as independent molecules has been found for the (2, 2-dimethylpropane-1, 3-diamine) chromium (III) complexes [34,35].

In the only other reported tris (curcuminoid) complex [36], the iron (III) metal center resides in a special position of the trigonal space group *P*-3, implying that the three curcuminoid ligands adopt the same conformation. In the In-complex **3**, all three theoretical conformations (“cis-up”, “cis-down”, and “trans”) are present simultaneously coordinated to the metal center (Figure 3), showing the great flexibility of the ligand. To the best of our knowledge, this is unique in two aspects: the first report of a crystal structure trapping three conformations for a ligand coordinated to a metal center and the first report of the observed “cis-up” conformation for the DiMeOC ligand.

### 3.2. NMR Solid State

The CP-MAS ^13^C NMR spectrum of the ligand (DiMeOC) reveals two singlet signals corresponding to the carbonyls of the β-diketone system. In the case of (DiMeOC)_2_-Ga, the carbonyls exhibit an overlapping single signal, attributed to the increased symmetry around the metal atom’s coordination plane. However, a distinct behavior is observed for the (DiMeOC)_3_-In complex, where three singlet signals are observed for the carbonyls of the three dicarbonyl systems. This observation indicates a correlation between the geometry observed in the SCDXR structures and the signals obtained for the carbonyls in solid-state NMR (Figure 4).

### 3.3. NMR Liquid State

The ^1^H NMR spectrum of DiMeOC (**1**) exhibits a broad singlet at 16.25 ppm for the OH proton and a singlet at 6.10 ppm for the methine proton. Both protons are involved in a strong intramolecular hydrogen bond, indicating the presence of the enol tautomer. The protons α to the diketone appear at 6.82 ppm, while the protons β to the diketone appear at 7.59 ppm, with a trans coupling constant of approximately 15.9 Hz. The methoxy groups are observed as singlets at 3.81 ppm and 3.83 ppm (Appendix A).

In the ^1^H NMR spectrum of (DiMeOC)_2_-Ga (**2**), the singlet at 6.04 ppm corresponds to the methine proton. The protons α to the diketone appear at 6.81 ppm, and the protons β to the diketone appear at 7.46 ppm, with a trans coupling constant of around 15.7 Hz. The methoxy groups are observed as singlets at 3.77 ppm and 3.78 ppm (Appendix A).

Similarly, in the ^1^H NMR spectrum of (DiMeOC)_3_-In (**3**), a singlet at 5.97 ppm corresponds to the methine proton. The protons α to the diketone appear at 6.84 ppm, and the protons β to the diketone appear at 7.49 ppm, with a trans coupling constant of approximately 15.6 Hz. The methoxy groups are observed as singlets at 3.77 ppm and 3.78 ppm (Appendix A).

The chemical shift toward lower frequencies observed in the methine and vinyl group protons of the metal complexes can be attributed to the presence of DMSO. This highly coordinating molecule is part of the complex during the NMR experiment [19]. DMSO contributes with electronic density, leading to a shielding effect in the coordination region. This phenomenon is described in Table 1 [9,37].

### 3.4. Optical Properties

The absorption and fluorescence spectra of compounds **1**–**3** were recorded in DMSO at room temperature. The spectra of compounds **1**–**3** exhibit an absorption band at 404, 392, and 355 nm attributed to π-π* transition (Figure 5). The charge transfer (CT) bands of compounds **2** and **3** at 473 and 421 nm demonstrate the formation of the complexes (Table 2) [38]. The maximum fluorescence emission of compounds **1**–**3** was observed near the Near-infrared (NIR) region [39]. Compounds **2** and **3** showed a low quantum yield compared to the ligands (Table 2). In the study of the optical properties of three different curcuminoids with Gallium, Asti et al. [9] suggested that metal ions induce a quenching of the fluorescence emission due to metallic complexation, which coincides with what has been obtained experimentally. Other authors attribute the decrease in the quantum yield of complexes of Ga and In to a charge transfer from the ligand to the metal [40,41].

### 3.5. IR Spectroscopy

The comparison between the IR spectra of the free ligand (dimethoxycurcumin) and the gallium and indium complexes showed the deprotonation of the enol with the disappearance of the band at 1682 cm^−1^ (stretch C=O) and the appearance of a new band at 1498 cm^−1^ for the Gallium complex and at 1495 cm^−1^ for the Indium complex (C=O in the coordinated ligand) [42]. The IR spectra of both complexes showed two new bands in the fingerprint region: at 458 cm^−1^ for the Gallium complex, and 453 cm^−1^ for the Indium complex. This reflects the coordination of the ligand with the metallic center (M-O) and agrees well with previous reports [43].

### 3.6. Stability Studies

The stability of the compounds, followed by UV-Vis and NMR techniques, showed no significant spectral changes over five days (see Supplementary Material). In these cases, both ligands and their complexes remained unchanged for 48 h, which agrees well with studies on other curcuminoids previously reported [44].

### 3.7. Cytotoxic Activity in Human Tumor Cells

The cytotoxic potential of DiMeOC has been reported to be even better than curcumin in in vitro models [38,39]. In these results, the DiMeOC ligand and its complexes exhibited significantly lower IC_50_ values compared to curcumin, a known effective agent against various cancer cell lines.

The biological properties of gallium and indium have been studied and associated with their ionic radii, compared to other essential biological ions such as Fe^3+^, Ca^2+^, and Na^+^ [7,45]. The higher potency and selectivity exerted by the gallium complex can be associated with the observed binding constant values between transferrin and Ga^3+^ (19.759 ± 0.25), resembling that of transferrin-Fe^3+^ (21.91) [46]. A less efficient cell absorption of In(III)-transferrin would explain its lesser cytotoxic activity [47].

(DiMeOC)_2_-Ga demonstrated exceptional selectivity among the complexes, displaying remarkable potential against the K562 cell line. Its potency was found to be four times higher than that of cisplatin against the K562 cell line and comparable to that observed for U251 and PC-3 cell lines. Notably (DiMeOC)_2_-Ga exhibited up to three times lower toxicity than cisplatin. These findings suggest that (DiMeOC)_2_-Ga is a promising therapeutic agent, with gallium seeming to provide greater potency and selectivity in the cancer lines; this has been confirmed in studies of gallium metallodrugs as promising treatments for various types of cancer [48,49]. In fact, the curcumin-gallium and diacetylcurcumin-gallium complexes have been reported to have promising antioxidant and cytotoxic activities on prostate, breast, and bladder cancer cell lines [50,51].

Furthermore, remarkable selectivity is observed when comparing the IC_50_ values in non-cancerous COS-7 cells to those of cancerous U251, PC-3, and K562 cell lines, with these values being 4, 2, and 11 times less cytotoxic, respectively. This selectivity is particularly noteworthy when compared to cisplatin (Table 3).

## 4. Materials and Methods

All the chemicals utilized in this study were commercially available, and prior to use, the solvents underwent purification using conventional methods.

Single-crystal X-ray diffractions (XDR) were performed on a Bruker diffractometer, specifically the Smart Apex model, equipped with Mo radiation (λ = 0.71073 Å), a CCD two-dimensional detector, and a low-temperature device. Data collection and reduction were carried out using APEX and SAINT-Plus programs. The structures were solved using direct methods via SHELXS-2013 software (Dept. of Structural Chemistry, Universität Göetingen, Germany) and refined using the Full-matrix least-squares procedure on *F^2^* with the SHELXL-2019 program [44]. CP/MASS ^13^C NMR spectra were recorded on a JEOL 600 MHz spectrometer (JEOL Mexico SA. de CV, CDMX) (15.0 kHz of MAS) using adamantane as the reference (298 K). ^1^H and ^13^C NMR liquid spectra were acquired in dimethyl sulfoxide (DMSO-*d_6_*) on a Bruker Fourier 400 MHz spectrometer (Billerica, MA, USA) and a Varian Unity Inova 500 MHz spectrometer (NMR Instruments, Palo Alto, CA, USA), respectively, with TMS serving as the internal reference. UV-Visible measurements were carried out using a Shimadzu U160 spectrophotometer (Spectralab Sceintific Inc., Markham, ON, Canada). Fluorescence experiments were measured on a FluoroMax spectrofluorimeter from HORIBA Scientific (HORIBA Instruments Inc., Irving, CA, USA). IR absorption spectra were recorded in the 4000–230 cm^−1^ range as KBr pellets on a BRUKER Tensor 27 spectrophotometer (Billerica, MA, USA).

Samples were dissolved in water as recommended in [44]. Thus, a suspension of 5 mg of the ligand (DiMeOC) and the Gallium and Indium complexes was prepared in distilled water; then, each sample was filtered off through a sintered filter to remove insoluble material. UV-Vis and ^1^H-NMR spectra were recorded in a Hewlett Packard 5484 UV-Vis Spectrophotometer and 400 MHz Bruker equipment, respectively. The structural stability of the compounds was monitored by UV-Vis for five days. Samples dissolved in DMSO-d6 for NMR determinations (time 1) were observed for possible structural changes for 48 h to match the incubation period of cells in the biological essays (time 2).

Mass spectra were obtained using a JEOL SX 102 A spectrometer (JEOL Mexico, SA de CV. CDMX) equipped with MALDI-Flight time technology. Melting points were determined using the electrothermal engineering IA9100X1 melting point apparatus, and the reported values are uncorrected.

The cytotoxicity of all compounds was assessed against several cancer cell lines, namely U251 (human glioblastoma), PC-3 (human Caucasian prostate adenocarcinoma), K562 (human Caucasian chronic myelogenous leukemia), HCT-15 (human colon adenocarcinoma), MCF-7 (human mammary adenocarcinoma), SKLU-1 (human lung adenocarcinoma), and COS-7 (non-tumoral monkey kidney). These cell lines were obtained from the U.S. National Cancer Institute (NCI) (Montgomery County, MD, USA). The cells were cultured in RPMI-1640 medium supplemented with 10% fetal bovine serum, 2 mM L-glutamine, 10,000 units/mL penicillin G sodium, 10,000 μg/mL streptomycin sulfate, 25 μg/mL amphotericin B (Invitrogen/Gibco^TM^, Thermo Fisher Scientific, Waltham, MA, USA), and 1% non-essential amino acids (Gibco). They were maintained at a temperature of 37 °C in a humidified atmosphere containing 5% CO_2_. The experiment’s cell viability exceeded 95%, as determined using trypan blue staining. To evaluate the cytotoxicity on human tumor cells, the protein-binding dye sulforhodamine B (SRB) was employed in a microculture assay, following the protocols established by the NCI [52,53,54]. The results were reported as IC_50_ values, representing the inhibitory concentration required for 50% inhibition. The IC_50_ values were determined using the Monks protocol [52], which involved plotting dose-response curves for each compound and estimating the concentration (IC_50_) at which a 50% inhibition was observed through non-linear regression analysis.

### Synthetic Procedures

Compound **1**. 4 g of curcumin in 120 mL of anhydrous acetone were reacted with 0.75 g of potassium carbonate (K_2_CO_3_) and 1.2 mL of dimethyl sulphate (SO_2_(OCH_3_)_2_) and refluxed with stirring for 48 h until the disappearance of the starting material. The solvent was removed under reduced pressure, and the product was extracted with a 3:7 mixture of AcOEt-H_2_O and NaOH 10% until SO_2_(OCH_3_)_2_ was removed from the organic phase. The product was purified by SiO_2_ column chromatography eluting with a mixture of hexane-CH_2_Cl_2_-methanol (MeOH) 50:45:5, and the product was dried under high vacuum (3), 63.2% yield. The product was recrystallized in MeOH, 65.4% yield. ^1^H NMR (500 MHz DMSO-*d_6_*): δ 3.81 (s, 6H,) 3.83 (s, 6H), 6.10 (s, 1H), 6.82 (d, 2H, *J* 15.84 H_vinyl_), 7.01 (d, 2H, *J* 8.42 H_aryl_), 7.26 (dd, 2H, *J* 8.42, 1.92 H_aryl_), 7.35 (d, 2H, *J* 1.92 H_aryl_), 7.59 (d, 2H, *J* 15.93 H_vinyl_), 16.25 (br s, 1H) ppm. ^13^C NMR (^13^C 1H 125 MHz, DMSO-*d_6_*): δ 55.57 (C-H), 100.99 (C-H), 110.47 (C_aryl_), 111.67 (C_aryl_), 122.03 (C_aryl_), 122.90 (C_vinyl_), 127.56 (C_aryl_), 140.40 (C_vinyl_), 149.02 (C_aryl_), 150.96 (C_aryl_), 183.19 (C=O) ppm. IR 3005 cm^−1^, 2926 cm^−1^, 2831 cm^−1^, 1624 cm^−1^, 1583 cm^−1^, 1504 cm^−1^, 1134 cm^−1^, 802 cm^−1^, 607 cm^−1^, 559 cm^−1^, 544 cm^−1^, 469 cm^−1^. LRMS: M+ 396.770. m.p. of orange crystal: 133.5 °C. T. C:69.68 H:6.10; E. C:69.98 H:6.31 (see Appendix A).

Compound **2**. 1 mmol of DiMeOC in 25 mL ethyl acetate reacted with 1 mmol of triethylamine (TEA), and later, a solution of gallium chloride (0.5 mmol) in methanol was added slowly. After 24 h of stirring at room temperature, an orange powder was formed in the flask, which was filtered and crystallized in DMF, 81.5% yield. ^1^H NMR (400 MHz DMSO-*d_6_*): δ 3.77 (s, 6H,) 3.78 (s, 6H), 6.04 (s, 1H), 6.81 (d, 2H, *J* 15.85 H_vinyl_), 6.96 (d, 2H, *J* 8.46 H_aryl_), 7.16 (d, 2H, *J* 8.46 H_aryl_), 7.27 (s, 2H, *J* 1.57 H_aryl_), 7.46 (d, 2H, *J* 15.58 H_vinyl_) ppm. ^13^C NMR (^13^C 1H 100 MHz, DMSO-*d_6_*): δ 55.46 (C-H), 55.53 (C-H), 102.0 (C-H), 110.14 (C_aryl_), 111.65 (C_aryl_), 122.47 (C_aryl_), 125.71 (C_vinyl_), 127.80 (C_aryl_), 139.48 (C_vinyl_), 148.98 (C_aryl_), 150.58 (C_aryl_), 183.20 (C=O) ppm. IR 1597 cm^−1^, 1578 cm^−1^, 1498 cm^−1^, 1259 cm^−1^, 1018 cm^−1^, 801 cm^−1^, 458 cm^−1^. HRMS observed: 859.2254; estimated: 859.2245. m.p. (red powder) 282.5 °C. T. C:64.20 H:5.39; E. C:65.17 H:5.39 (see Appendix A).

Compound **3**. 1 mmol of DiMeOC in 25 mL ethyl acetate reacted with 1 mmol of triethylamine (TEA), and later, a solution of indium nitrate (0.5 mmol) in methanol was added slowly. After 24 h of stirring at room temperature, an orange powder was formed in the flask, which was filtered off and crystallized in DMF, 66.3% yield. ^1^H NMR (400 MHz DMSO-*d_6_*): δ 3.77 (s, 6H,) 3.78 (s, 6H), 5.97 (s, 1H), 6.84 (d, 2H, *J* 15.68 H_vinyl_), 6.97 (d, 2H, *J* 8.50 H_aryl_), 7.19 (dd, 2H, *J* 8.45, 1.93 H_aryl_), 7.28 (s, 2H, *J* 1.95 H_aryl_), 7.49 (d, 2H, *J* 15.59 H_vinyl_) ppm. ^13^C NMR (^13^C 1H 100 MHz, DMSO-*d_6_*): δ 55.45 (C-H), 55.54 (C-H), 103.51 (C-H), 110.29 (C_aryl_), 111.67 (C_aryl_), 122.46 (C_aryl_), 126.30 (C_vinyl_), 127.78 (C_aryl_), 139.76 (C_vinyl_), 148.97 (C_aryl_), 150.63 (C_aryl_), 185.13 (C=O) ppm. IR 1632 cm^−1^, 1579 cm^−1^, 1495 cm^−1^, 1254 cm^−1^, 967 cm^−1^, 804 cm^−1^, 453 cm^−1^. HRMS observed: 1301.3638; estimated: 1301.3601. m.p. of yellow powder 235.5 °C. T. C:63.70 H:5.35; E. C:63.33 H:5.23 (see Appendix A).

## 5. Conclusions

We successfully describe the synthesis of two new homoleptic metal complexes (Galium and Indium, Family 13) with dimethoxycurcumin. Their structures were unambiguously determined by the single crystal X-ray technique. The solid-state analysis allowed us to assign the metal-ligand relationship as ML_2_ for Gallium and ML_3_ for Indium.

The crystal structures of these complexes demonstrate octahedral geometries for both the gallium and indium compounds. The discovery of three distinct conformations of the same ligand within a molecule is of particular significance in the case of (DiMeOC)_3_-In. To our knowledge, this represents the first reported instance of such a structure in literature.

Biological assays of these metal complexes against non-cancer cell lines (COS-7) showed that (DiMeOC)_2_-Ga was less toxic (IC_50_ = 23.2 ± 0.19 mM) than dimethoxycurcumin (IC_50_ = 2.02 ± 0.7 mM) and Cisplatin (IC_50_ = 7.2 ± 0.6 mM). The cell death in the K-562 lines with the Gallium complex was better (IC_50_ = 2.0 ± 0.2 mM) in relation to cisplatin (IC_50_ = 8.6 ± 0.9 mM), making it a promising treatment for leukemia. This remarkable selectivity demonstrates the potential of (DiMeOC)_2_-Ga as a highly effective therapeutic agent.

In addition, our experimental findings may become a useful contribution when comparisons of structural features and biological properties are explored theoretically.

This constitutes the first report of crystal structures of indium and gallium complexes of curcuminoid type, and their therapeutic potential is confirmed while adding new molecular resources to the worldwide battle against cancer.

## Figures and Tables

**Figure 1 ijms-24-16324-f001:**
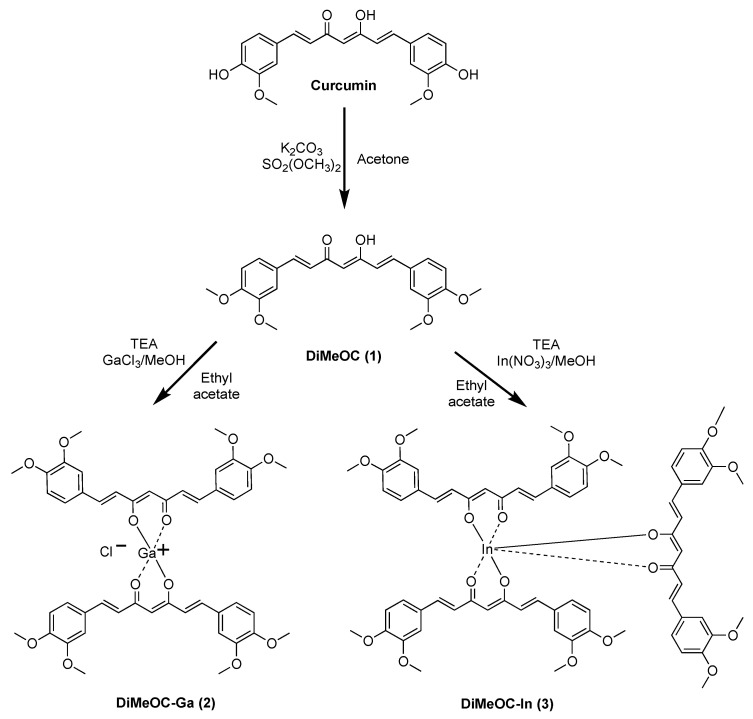
Curcuminoids and their gallium and indium complexes.

**Figure 2 ijms-24-16324-f002:**
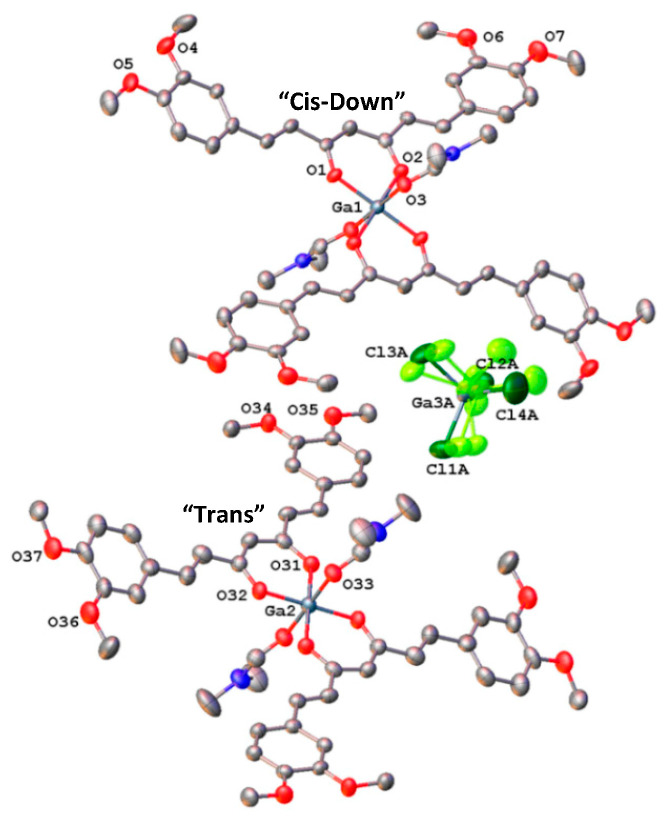
A view of the cations of [*trans*-(DiMeOC)_2_-Ga•(DMF)_2_^+^] (**2**), showing the atom-labelling scheme. Displacement ellipsoids are drawn at the 50% probability.

**Figure 3 ijms-24-16324-f003:**
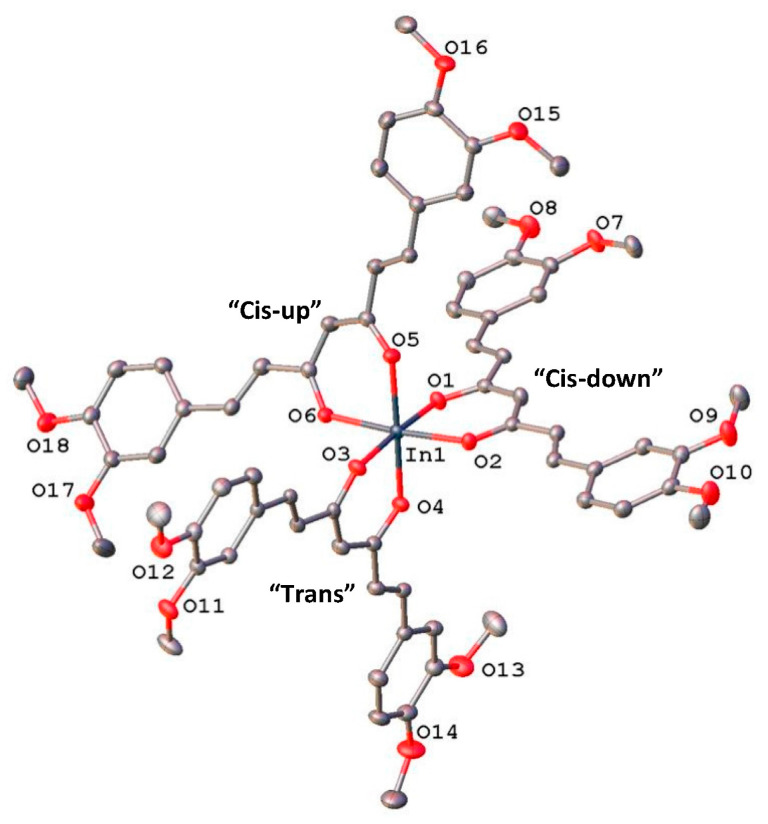
A view of compound **3**, showing the atom-labeling scheme and the three ligand conformations. Displacement ellipsoids are drawn at the 50% probability.

**Figure 4 ijms-24-16324-f004:**
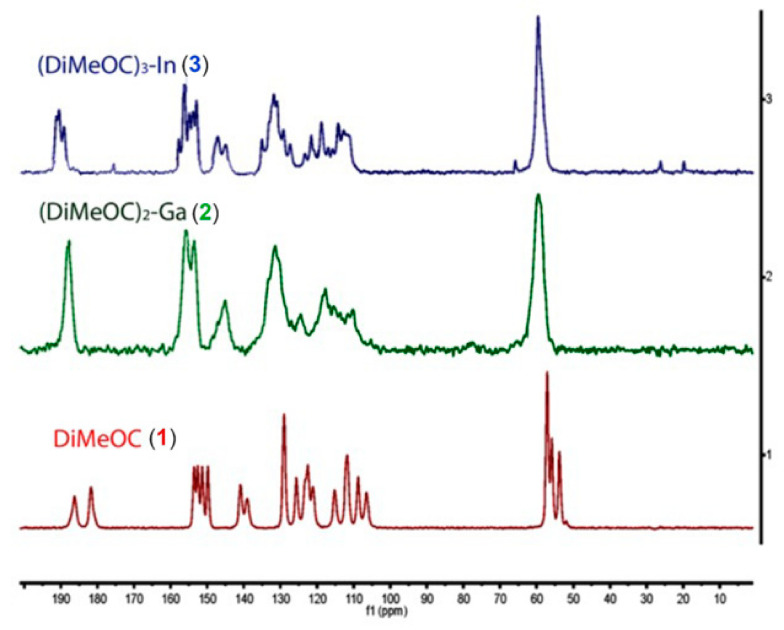
150 MHz ^13^C ssNMR spectra of DiMeOC and its (DiMeOC)_2_-Ga and (DiMeOC)_3_-In complexes.

**Figure 5 ijms-24-16324-f005:**
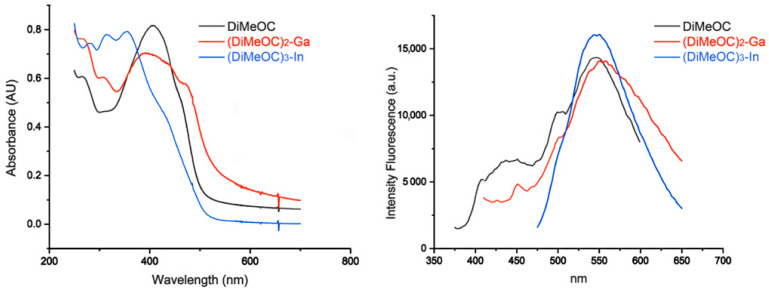
Absorption and emission spectra of DiMeOC and its (DiMeOC)_2_-Ga and (DiMeOC)_3_-In complexes.

**Table 1 ijms-24-16324-t001:** ^1^H NMR chemical shifts of the methine proton of ligand and its Gallium e Indium complexes.

Compounds	δ (ppm) of Methine Proton in DMSO-*d_6_*
DiMeOC	6.10
(DiMeOC)_2_-Ga	6.04
(DiMeOC)_3_-In	5.97

**Table 2 ijms-24-16324-t002:** Optical properties of compounds **1**–**3**.

Compounds	CT	ε_max_ (L/cm mol)	π-π*	ε_max_ (L/cm mol)	Abs lmax	IF lmax	Φ ^a^
DiMeOC	-	-	404	16,253.63	404	549	0.01558
(DiMeOC)_2_-Ga	473	10,327.02	392	12,048.19	392	558	0.00524
(DiMeOC)_3_-In	421	18,866.11	355	29,274.99	356	547	0.00498

^a^ Standard reference of quinine sulphate.

**Table 3 ijms-24-16324-t003:** IC_50_ (μM) for cancer cell lines with curcumin, compounds **1**–**3**, and cisplatin.

Compounds	U251	PC-3	K562	HCT15	MCF7	SKLU-1	COS-7
Curcumin	20.5 ± 1.7	22.5 ± 1.9	16.4 ± 0.04	21.3 ± 0.9	18.4 ± 0.9	12.9 ± 0.7	21.8 ± 2.1
DiMeOC	2.02 ± 0.07	5.9 ± 0.6	6.5 ± 1.4	6.6 ± 0.3	6.9 ± 1.0	7.2 ± 0.4	2.02 ± 0.07
(DiMeOC)_2_-Ga	5.3 ± 0.1	10.5 ± 0.3	2.0 ± 0.2	-	15.5 ± 0.4	-	23.2 ± 1.9
(DiMeOC)_3_-In	6.3 ± 0.6	7.6 ± 0.05	-	7.9 ± 0.2	9.7 ± 0.4	6.2 ± 0.4	-
Cisplatin	4.7 ± 0.4	8.94 ± 0.9	8.6 ± 0.9	10.0 ± 0.9	9.4 ± 1.0	4.3 ± 0.5	7.2 ± 0.6

## Data Availability

Data are contained within the article and Appendix A. The following are available online, CCDC-2250194 Compound **2**, CCDC-2250195 Compound **3** contain the supplementary crystallographic data for this paper. These data can be obtained free of charge via https://www.ccdc.cam.ac.uk/structures/ accessed on 11 November 2023, The Cambridge Crystallographic Data Centre, 12 Union Road, Cambridge CB2 1EZ, UK, fax: +44(0) 1223-336033.

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
