# Peer review of "First Gallium and Indium Crystal Structures of Curcuminoid Homoleptic Complexes: All-Different Ligand Stereochemistry and Cytotoxic Potential"

_ijms, 2023, doi:10.3390/ijms242216324_

Round 1
Reviewer 1 Report
Comments and Suggestions for Authors This article is reported the Gallium and Indium crystal structures with curcuminoid.The cytoxicity studies shows decrease in cytotoxicity of the ligand DiMeOC. Only again K562 cells it has better activity. The author need to explain this selectivity.
Also it will be good to evaluate the stability of the complexes and ligand in solution phase.
Author Response
Please check word document

Reviewer 2 Report
Comments and Suggestions for Authors
In this paper, the author has explored the metal complexation of dimethoxycurcumin, which possesses a β-diketone system like acetylacetone (ACAC), and developed appropriate conditions for the synthesis of the gallium and indium metal complexes using salts of both ions. The focus of this paper mainly examines the relationship between the ligand and metal coordination number, primarily through solid state analysis. The novelty of the discussion is missing, and a detailed comparison is needed. Many sentences have not been clearly explained. But the topic is interesting for the referee. Therefore, I recommend publication after major revisions.
1. The abstract in this article is not so exciting/good; the author needs to rewrite the abstract and needs to include the motivation and necessity of this research.
2. I think that the introduction is not sufficient, and the introduction should be elaborated with state of art. Needs to include an introduction to are Advantages and Disadvantages and the importance of this research and needs to explain the importance of this research.
3. I think that the author needs to include appropriate working principles/applications for this paper for better understanding.
4. Page No:6; Table.1 title, the author needs to correct.
5. Page No: 7; Optical properties: I think that a detailed explanation is needed. And include the UV-Vis and Emission spectra in the main paper (not in supplementary) for better understanding.
6. Page No: 7; Optical properties: The author has written “The decrease in quantum yield for complexes of Ga and In can be explained by the charge transfer from the ligand to the metal.” Need detailed explanation about charge transfer from the ligand to the metal. Needs to include lifetime experiments for further clarification.
7. Table 2: what is the standard reference for quantum yield, that needs to be included?
8. In the result and discussion, the author needs to include a comparison of the present work with previous work. And needs to explain the importance of the present work.
9. The Discussion/conclusions of this paper do not show all the results. I think that the author needs to rewrite/elaborate the conclusions.
10. Please verify each sentence of this paper, Some typo errors.
Comments on the Quality of English LanguageModerate editing of the English language is required. Please verify each sentence of this paper, Some typo errors.
Author Response
Please see document.

Round 2
Reviewer 2 Report
Comments and Suggestions for Authors
Thank you to the author for addressing all the raised comments from reviewers. The authors have carried out the modifications and the corrections to the queries raised and indeed have provided more information in the revised version. Therefore, I recommend this manuscript for publication in IJMS as it is without further modifications.
Comments on the Quality of English Language
Minor editing of English language required